Dynamics of the hindgut microbiota of the Japanese honey bees (Apis cerana japonica) throughout the overwintering period

http://orcid.org/0000-0002-3179-2106 Suzuki Akihiko 1 suzuki.akihiko@nies.go.jp
http://orcid.org/0000-0002-7855-7621 Hisamoto Shumpei 2
Sakamoto Yoshiko 1
1 National Institute for Environmental Studies , Tsukuba, Ibaraki , Japan
2 Meiji Institute for Advanced Study of Mathematical Sciences, Meiji University , Nakano, Tokyo , Japan
Beddoe Travis
Electronic publication date: 2025 Oct 13
Publication date: 2025
Volume: 13
Electronic Location ID: e20050
Received 2025 Jul 9; Accepted 2025 Aug 17
Copyright: © 2025 Suzuki et al.
Copyright year: 2025
Copyright holder: Suzuki et al.
License: This is an open access article distributed under the terms of the Creative Commons Attribution License, which permits unrestricted use, distribution, reproduction and adaptation in any medium and for any purpose provided that it is properly attributed. For attribution, the original author(s), title, publication source (PeerJ) and either DOI or URL of the article must be cited.
License URL: https://creativecommons.org/licenses/by/4.0/

Keywords: Hindgut microbiota, Honey bees, Apis cerana japonica, Overwintering

Funding: Grant-in-Aid for Early-Career Scientists 23K13972 AS from the Japan Society for the Promotion of Science This study was supported by a Grant-in-Aid for Early-Career Scientists (23K13972, AS) from the Japan Society for the Promotion of Science. The funders had no role in study design, data collection and analysis, decision to publish, or preparation of the manuscript.

==============================
Honey bees play crucial roles as pollinators in natural, agricultural, and ecological systems. The role of gut microbiota in the overwinter survival of honey bees is gaining attention. Compared with Western honey bees (Apis mellifera), Eastern honey bees (Apis cerana) are more tolerant to low-temperature stress. This study compared the hindgut microbiota of the Japanese honey bees (Apis cerana japonica), a subspecies of A. cerana, during the overwintering period (December) with that before overwintering (October) and after overwintering (March) to estimate beneficial hindgut bacteria contributing to survival during the overwintering period. Overall, the hindgut microbiota of A. c. japonica was occupied by Actinobacteriota, Bacteroidota, Firmicutes, and Proteobacteria at the phylum level and Apibacter, Bifidobacterium, Bombilactobacillus, Gilliamella, Lactobacillus, and Snodgrassella at the genus level. The hindgut microbiota composition of A. c. japonica was similar to that of A. cerana in other regions, suggesting that phylogeny influenced the composition. Many sequences assigned to these six core genera showed <98.7% similarity to type strains, indicating potential novel bacterial species. The relative abundance of Bifidobacterium, Bombilactobacillus, and Lactobacillus was higher during overwintering than in other periods. Our findings highlight changes in the core bacteria of the hindgut microbiota of A. c. japonica during overwintering and also suggest the presence of novel candidate bacterial species. The roles of the bacteria that were increased during the overwintering period require further elucidation.

Introduction

The gut microbiota of honey bees plays a critical role in their health (Raymann & Moran, 2018), including decomposing dietary compounds (Engel, Martinson & Moran, 2012), producing short-chain fatty acids (SCFAs) as an energy source (Zheng et al., 2017), degrading potentially toxic plant metabolites (Motta et al., 2022), inhibiting the growth of honey bee pathogens (Wu et al., 2014), and stimulating the immune system (Kwong, Mancenido & Moran, 2017; Motta & Moran, 2024). Disruption of the gut microbiota composition due to antibiotic treatment and pesticide exposure causes dysbiosis, leading to host mortality (Raymann, Shaffer & Moran, 2017; Motta & Moran, 2024).

During winter, honey bees survive the severe cold environment in a metabolically and physically active state that is essential for ensuring the colony’s survival until the following spring (Moeller, 1977; Doeke, Frazier & Grozinger, 2015). Cold stress is a major cause of individual and colony mortality in honey bees and also increases the risk of disease and infection outbreaks (Doeke, Frazier & Grozinger, 2015; Xu et al., 2017). During overwintering, feeding is essentially limited to food stored within the colony (pollen, bee bread, and honey). To cope with the surrounding cold stress, the honey bees must maintain the temperature of the colony’s outer edge and core by vigorously vibrating their flight muscles to generate heat (Doeke, Frazier & Grozinger, 2015). The gut microbiota of overwintering honey bees has been increasingly recognized for its beneficial role in survival during the overwintering period. Most studies have focused on the Western honey bees (Apis mellifera), reporting an increase in gut bacteria abundance during this period (Kešnerová et al., 2020) along with changes in gut microbiota composition (Bleau et al., 2020; Kešnerová et al., 2020; Liu et al., 2021; Castelli et al., 2022; Li et al., 2022; Brar et al., 2025). These findings suggest that gut bacteria may play crucial roles in energy absorption and immune function, thereby substantially contributing to survival during the overwintering period.

The Apis genus is naturally distributed across Asia, Europe, and Africa (Ji, 2021). The Western honey bees are widely distributed worldwide, including their native regions of Europe, Africa, and the Middle East, while Eastern honey bees (Apis cerana) are found throughout South, Southeast, and East Asia (Ji, 2021). Compared to Western honey bees, Eastern honey bees exhibit superior cold tolerance and are better adapted to surviving the harsh overwintering period (Li et al., 2012; Xu et al., 2017). In consideration of the advantageous role of the gut microbiota in the overwintering process of honey bees, we hypothesized that the gut microbiota plays a significant role in enabling Eastern honey bees to tolerate cold stress and successfully overwinter. Characterizing the hindgut microbiota of overwintering honey bees will help narrow down the candidate bacteria beneficial for survival under cold and harsh environments, providing novel insights into the symbiotic relationships between honey bees and their hindgut microbiota.

This study focused on the Japanese honey bees (Apis cerana japonica), a subspecies of the Eastern honey bee that is native to Japan. The aim was to develop a comprehensive inventory of the hindgut microbiota of A. c. japonica using high-throughput sequencing of the V3–V4 region of the bacterial 16S rRNA gene. In addition, we compared the composition of the hindgut microbiota before, during, and after the overwintering period to identify distinctive microbial features associated with successful overwintering.

Materials and Methods

Sample collection

The study samples were collected from four A. c. japonica colonies in Ibaraki, Japan. Two colonies were maintained by our laboratory at the National Institute of Environmental Studies in Tsukuba City, while the other two were managed by local beekeepers in Tsukuba City and Inashiki District, respectively. We sampled 30 foragers from each colony using a net with clean plastic cups over three periods: October 2022 (before overwintering, BO group), December 2022 (during overwintering, OW group), and March 2023 (after overwintering, AO group) (Table 1). The total number of samples was 360 (30 honey bees per colony × four colonies × three periods). All samples were immediately placed on ice after collection and stored at −80 °C until DNA extraction.

Table 1 Detailed information on sampling in this study.

Period1)	Sampling date	Colony ID	Location2)	DNA sample ID	
BO	10/19/2022	H	Tsukuba city, Ibaraki, Japan	H10-1, H10-2, H10-3	
	10/27/2022	I	Inashiki district, Ibaraki, Japan	I10-1, I10-2, I10-3	
	10/20/2022	T	NIES	T10-1, T10-2, T10-3	
	10/20/2022	X	NIES	X10-1, X10-2, X10-3	
OW	12/19/2022	H	Tsukuba city, Ibaraki, Japan	H12-1, H12-2. H12-3	
	12/18/2022	I	Inashiki district, Ibaraki, Japan	I12-1, I12-2. I12-3	
	12/19/2022	T	NIES	T12-1, T12-2. T12-3	
	12/19/2022	X	NIES	X12-1, X12-2. X12-3	
AO	3/15/2023	H	Tsukuba city, Ibaraki, Japan	H3-1, H3-2, H3-3	
	3/16/2023	I	Inashiki district, Ibaraki, Japan	I3-1, I3-2, I3-3	
	3/7/2023	T	NIES	T3-1, T3-2, T3-3	
	3/7/2023	X	NIES	X3-1, X3-2, X3-3	
Notes:

1) BO, Before overwintering; OW, during overwintering; AO, after overwintering.

2) NIES, National Institute for Environmental Studies (Tsukuba, Ibaraki, Japan).

DNA extraction

After thawing the honey bees on ice, they were surface-sterilized by immersing them in 70% ethanol for 30 s, followed by rinsing with ultrapure water for 30 s. The hindguts, including the pylorus, ileum, and rectum, were carefully removed on ice using sterile forceps. Ten hindguts were pooled into 2.0 mL sterile tubes containing TE buffer (10 mmol L−1 Tris-HCl and 1 mmol L−1 EDTA-2Na, pH 8.0) supplemented with 5% (v/v) Triton X-100 (MP Biomedicals, Irvine, CA, USA) and glass beads (1.0 mm diameter). The hindguts were disrupted by three cycles of crushing at 3,200 rpm for 30 s using Beads Crusher µT-12 (Taitec, Saitama, Japan) and 30 s of cooling on ice. The homogenates were centrifuged at 6,000× g for 10 min to sediment debris. Total bacterial DNA was extracted from 180 μL of the resulting supernatant using the DNeasy Blood and Tissue Kit (Qiagen, Hilden, Germany) according to the manufacture’s instructions. The DNA concentration of the 36 samples (three replicates per colony × four colonies × three sampling periods) was measured using a NanoDrop One spectrophotometer (Thermo Fisher Scientific, Waltham, MA, USA).

High-throughput sequencing of the V3–V4 region of the bacterial 16S rRNA gene

The bacterial V3–V4 hypervariable regions of the 16S rRNA gene were amplified using a universal primer set comprising 341F (5′-ACACTCTTTCCCTACACGACGCTCTTCCGATCT-NNNNN-CCTACGGGNGGCWGCAG-3′) and 805R (5′-GTGACTGGAGTTCAGACGTGTGCTCTTCCGATCT-NNNNN-GACTACHVGGGTATCTAATCC-3′). These primers include adapter sequences compatible with the Illumina library preparation workflow and are specific to the amplification of the bacterial V3–V4 regions. Polymerase chain reaction (PCR) was performed using template DNA (5 ng µl−1) with Blend Taq Plus polymerase (Toyobo, Osaka, Japan) following the manufacturer’s instructions. The PCR cycling conditions were initial denaturation at 94 °C for 2 min, followed by 30 cycles of denaturation at 94 °C for 30 s, annealing at 55 °C for 30 s, and extension at 72 °C for 30 s. The resulting PCR amplicons were verified by 1.5% agarose gel electrophoresis at 100 V for 25 min, stained with ethidium bromide, and visualized under UV illumination. The PCR amplicons were sent to Bioengineering Lab. Co., Ltd. (Kanagawa, Japan) for sequencing. The amplicons were purified using AMPure XP Beads (Beckman Coulter, Brea, CA, USA), and the DNA concentrations were measured using a Synergy H1 multimode microplate reader (Agilent Technologies, Santa Clara, CA, USA) and a QuantiFluor dsDNA System (Promega, Madison, WI, USA). Library preparation was performed using the purified amplicons with dual-index barcoding to enable sample multiplexing. After determining the library concentrations as described above, quality checking was performed using a dsDNA 915 Reagent kit (Agilent Technologies) on a Fragment Analyzer System (Agilent Technologies). Each library was pooled at an equimolar concentration and underwent 300 bp paired-end sequencing using MiSeq Reagent Kit v3 (Illumina, San Diego, CA, USA) on a MiSeq benchtop sequencer (Illumina).

Data analysis

To generate amplicon sequence variants (ASVs), DADA2 ver. 1.16 (Callahan et al., 2016) in RStudio software ver. 2023.12.0 + 369 was used for trimming and filtering the paired-end FASTQ reads data obtained following MiSeq high-throughput sequencing. Low-quality distributions and primers from each forward and reverse sequence were trimmed using parameters set to truncLen = c(290, 230) and trimLeft = c(17,21), respectively, and then the sequences were filtered using maxN = 0, maxEE = c(2,2), and truncQ = 2 parameters. Next, the forward and reverse reads were merged, after which chimeras and short reads (<400 bp) were discarded. The taxonomic classification of representative ASVs from phylum to genus level was assigned using the SILVA ver. 138.1 prokaryotic SSU database (Quast et al., 2013) as a reference dataset. Before downstream analysis, ASVs that were unclassified at the phylum level and were assigned as non-bacteria groups (e.g., chloroplasts and mitochondria) were manually removed.

To standardize sequencing depth at minimum read count across samples, abundance-based read resampling was performed using the rrarefy function of the vegan package ver. 2.6.4 (Oksanen et al., 2023) among all samples. Rarefaction curves were generated using the rarecurve function of the vegan package and the ggplot2 package ver. 3.4.2 (Wickham, 2016). The Coverage function in the entropart package ver. 1.6.12 (Marcon & Hérault, 2015) was used to calculate coverage and evaluate whether the sequencing depth was sufficient to fully represent the bacterial communities in each sample before and after rarefaction. After pooling ASVs at the lowest taxonomic level (bacterial genus), the bacterial community composition at the phylum and genus levels was visualized for each sample as bar plots using ggplot2. Taxa with a relative abundance of <1% across all samples were grouped and represented as “Others”.

Bacterial genera with a relative abundance >1% across all samples were defined as the core hindgut bacteria of Apis cerana japonica. All ASVs assigned to these core genera underwent similarity searches against the bacterial 16S rRNA gene sequence database in EzBioCloud (Yoon et al., 2017) to identify the closest related species.

Non-metric multidimensional scaling (NMDS) plot was generated based on Bray–Curtis dissimilarity using the metaMDS function from the vegan package and visualized using the ggplot2 package, in order to assess the β-diversity of the hindgut microbiota at the ASV level across the three time periods.

Statistical analysis

To detect differences in the hindgut microbial compositions among the three periods, we performed pairwise comparisons using permutational multivariate analysis of variance (PERMANOVA) based on the Bray–Curtis dissimilarity index with 9,999 permutations using the pairwise.adonis function of the pairwiseAdonis package ver. 0.4.1 (Martinez, 2020).

To investigate the effect of sampling period on the abundance of core bacterial genera, a generalized linear mixed model (GLMM) analysis was conducted, assuming a Poisson distribution with a log link function, using the glmer function from the lme4 package ver. 1.1.32 (Bates et al., 2015). The read count of each core bacterial genus was set as the response variable, with sampling period as a fixed effect and colony as a random effect. P-values < 0.05 were considered statistically significant for all comparisons.

Results

Sequence dataset overview

The high-throughput sequencing generated 1,381,026 raw reads (mean ± SD: 38,362 ± 5,199) from all samples. After trimming and filtering, 1,098,689 high-quality reads (mean ± SD: 30,519 ± 3,678) remained (Table S1), clustering into 260 ASVs (mean ± SD: 59 ± 10). Next, all nonbacterial reads were removed from all samples, and a read count-based cutoff was applied to match the minimum read count (21,808), resulting in a final dataset of 241 ASVs (mean ± SD: 54 ± 8). The rarefaction curves for each sample plateaued at the minimum read depth (Fig. S1), and the estimated coverage was exceeded 99% for all samples (Table S2), indicating that the sequencing depth was sufficient to identify most of the hindgut bacteria in the study samples.

Hindgut microbiota composition

The composition at the phylum and genus levels for each period were described for those taxa with a relative abundance >1% across all samples, while those with a relative abundance <1% were grouped as “Others” (Fig. 1). At the phylum level, the hindgut microbiota of A. c. japonica was dominated by Actinobacteriota (0.4–13.1%), Bacteroidota (11.7–36.3%), Firmicutes (9.2–33.9%), and Proteobacteria (35.2–68.6%), collectively accounting for >99.9% of the relative abundance. At the genus level, six bacterial genera, namely Apibacter (11.6–36.3%), Bifidobacterium (0.2–13.1%), Bombilactobacillus (0.3–13.6%), Gilliamella (26.6–59.8%), Lactobacillus (6.9–28.2%), and Snodgrassella (0.7–22.4%) predominated, accounting for >96% of the hindgut microbiota in all three periods and all samples. The relative abundance of “Unclassified”, of which sequences were not assigned at the genus level, and “Others” was 0.0–5.4% and 0.0–12.1%, respectively. The details of relative abundance at the phylum and genus levels are listed in Tables S3–S6.

Figure 1 Hindgut microbiota composition of the Japanese honey bees (Apis cerana japonica) sampled for each period at the (A) phylum and (B) genus levels.

BO, before overwintering; OW, during overwintering; AO, after overwintering.

Further examination of the six major bacterial genera revealed that, except for genus Bifidobacterium, many ASVs assigned to the core genera exhibited sequence similarities with type strains in the EzBioCloud database (Yoon et al., 2017) below the 98.7% threshold, which is commonly used to distinguish closely related species (Chun et al., 2018) (Table 2). Notably, 90% (27/30) of the ASVs assigned to Gilliamella showed this pattern, followed by Snodgrassella (65%, 13/20), Bombilactobacillus (60%, 3/5), Apibacter (53.8%, 7/13), and Lactobacillus (41.6%, 5/12), suggesting potentially novel bacterial species within the hindgut microbiota of A. c. japonica.

Table 2 List of BLAST results against the EzBioCloud 16S rRNA database of ASVs assigned to the six core bacteria genera in the hindgut of the Japanese honey bees (Apis cerana japonica).

Assigned genus	ASV ID 1)	Length (bp)	Top-hit taxon (strain level) 2)	Accession ID	Similarity (%)	
Apibacter	ASV1	423	Apibacter sp. B3924	WINM01000002	100	
	ASV16	423	Apibacter mensalis R-53146T	LIVM01000008	99.5	
	Apibacter sp. B3924	WINM01000002	99.5	
	ASV9	423	Apibacter mensalis R-53146T	LIVM01000008	99.8	
	Apibacter sp. B3924	WINM01000002	99.8	
	ASV32	423	Apibacter sp. B3924	WINM01000002	99.8	
	ASV33	423	Apibacter mensalis R-53146T	LIVM01000008	100	
	ASV118	423	Apibacter sp. B3924	WINM01000002	96.2	
	ASV76	426	Apibacter sp. B3924	WINM01000002	97.9	
	ASV133	409	Apibacter mensalis R-53146 T	LIVM01000008	91.0	
	Apibacter sp. B3924	WINM01000002	91.0	
	ASV151	425	Apibacter sp. B3924	WINM01000002	97.6	
	ASV254	423	Apibacter sp. B3924	WINM01000002	90.6	
	ASV280	423	Apibacter sp. B3924	WINM01000002	91.0	
	ASV174	431	Apibacter mensalis R-53146 T	LIVM01000008	90.5	
	ASV279	422	Apibacter sp. B3924	WINM01000002	99.6	
Bifidobacterium	ASV51	408	Bifidobacterium indicum JCM 1302T	LC071807	100	
	ASV11	410	Bifidobacterium sp. 7101	AWUN01000009	100	
Bombilactobacillus	ASV34	432	Bombilactobacillus mellifer Bin4NT	JX099543	99.8	
	ASV47	431	Uncultured Firmicutes bacterium D08062C1	HM215046	98.1	
	ASV169	430	Uncultured Firmicutes bacterium D08062C1	HM215046	89.4	
	ASV12	429	Bombilactobacillus mellis Hon2T	KQ033880	100	
	ASV23	431	Uncultured Firmicutes bacterium D08062C1	HM215046	98.4	
Lactobacillus	ASV6	428	Lactobacillus panisapium Bb 2-3T	KX447147	100	
	ASV10	430	Lactobacillus panisapium Bb 2-3T	KX447147	99.8	
	ASV15	428	Lactobacillus panisapium Bb 2-3T	KX447147	99.5	
	ASV24	429	Lactobacillus melliventris Hma8NT	JX099551	99.5	
	ASV90	429	Lactobacillus panisapium Bb 2-3 T	KX447147	98.4	
	ASV135	427	Lactobacillus panisapium Bb 2-3T	KX447147	99.5	
	ASV164	428	Lactobacillus panisapium Bb 2-3 T	KX447147	95.6	
	ASV129	430	Lactobacillus huangpiensis F306-1T	LC597580	99.8	
	ASV267	429	Lactobacillus panisapium Bb 2-3T	KX447147	99.3	
	ASV271	427	Lactobacillus panisapium Bb 2-3 T	KX447147	96.7	
	ASV292	428	Lactobacillus panisapium Bb 2-3 T	KX447147	93.7	
	ASV185	428	Lactobacillus panisapium Bb 2-3 T	KX447147	97.9	
Gilliamella	ASV2	428	Gilliamella apicola wkB11	JFON01000004	97.4	
	ASV3	428	Gilliamella apis NO3T	NASD01000045	100	
	ASV4	428	Gilliamella apicola wkB7	CM004509	98.6	
	ASV5	428	Gilliamella apis NO3 T	NASD01000045	96.7	
	ASV8	428	Gilliamella apicola wkB11	JFON01000004	97.7	
	ASV13	430	Gilliamella apis NO3T	NASD01000045	99.8	
	ASV14	428	Gilliamella apicola wkB7	CM004509	98.4	
	ASV22	429	Gilliamella apicola wkB7	CM004509	98.4	
	ASV28	429	Gilliamella apicola wkB7	CM004509	98.4	
	ASV30	431	Gilliamella apicola wkB11	JFON01000004	97.5	
	ASV39	429	Gilliamella apicola wkB11	JFON01000004	97.4	
	ASV41	427	Gilliamella apicola wkB7	CM004509	98.4	
	ASV58	427	Gilliamella apicola App2-1	LZGR01000055	99.5	
	ASV71	428	Gilliamella apicola wkB7	CM004509	98.4	
	ASV73	429	Gilliamella apicola wkB11	JFON01000004	95.3	
	ASV96	428	Gilliamella apicola wkB11	JFON01000004	94.0	
	ASV100	432	Gilliamella bombi LMG 29879 T	FMWS01000047	95.4	
	ASV102	431	Gilliamella apicola wkB7	CM004509	96.5	
	ASV104	432	Gilliamella apis NO3 T	NASD01000045	97.7	
	ASV109	428	Gilliamella apicola wkB11	JFON01000004	96.5	
	ASV138	428	Gilliamella apicola wkB7	CM004509	96.0	
	ASV149	428	Gilliamella apicola wkB7	CM004509	96.0	
	ASV162	427	Gilliamella apicola wkB7	CM004509	98.1	
	ASV170	432	Gilliamella apicola wkB11	JFON01000004	95.4	
	ASV177	429	Gilliamella apicola wkB11	JFON01000004	97.2	
	ASV190	428	Gilliamella apicola wkB11	JFON01000004	93.7	
	ASV207	428	Gilliamella apicola wkB7	CM004509	96.3	
	ASV231	428	Gilliamella apicola wkB7	CM004509	96.3	
	ASV275	427	Gilliamella apicola wkB7	CM004509	97.7	
	ASV298	427	Gilliamella apicola wkB7	CM004509	96.5	
Snodgrassella	ASV7	428	Snodgrassella alvi wkB298	MEIK01000026	100	
	ASV17	428	Snodgrassella alvi wkB298	MEIK01000026	99.8	
	ASV18	428	Snodgrassella alvi wkB298	MEIK01000026	98.6	
	ASV21	429	Snodgrassella alvi wkB298	MEIK01000026	99.0	
	ASV25	429	Snodgrassella alvi wkB298	MEIK01000026	98.8	
	ASV27	430	Snodgrassella alvi WF3-3	MEIO01000062	98.6	
	ASV31	428	Snodgrassella alvi wkB298	MEIK01000026	98.8	
	ASV35	428	Snodgrassella alvi wkB298	MEIK01000026	98.4	
	ASV40	428	Snodgrassella gandavensis LMG 30236 T	OU943324	98.6	
	ASV45	428	Snodgrassella gandavensis LMG 30236T	OU943324	98.8	
	ASV60	428	Snodgrassella alvi wkB298	MEIK01000026	98.6	
	ASV66	430	Snodgrassella alvi WF3-3	MEIO01000026	98.8	
	ASV111	431	Snodgrassella alvi wkB298	MEIK01000026	98.4	
	ASV143	429	Snodgrassella alvi wkB298	MEIK01000026	96.5	
	ASV234	431	Snodgrassella alvi wkB298	MEIK01000026	98.1	
	ASV235	429	Snodgrassella alvi wkB298	MEIK01000026	96.3	
	ASV276	433	Snodgrassella alvi wkB298	MEIK01000026	93.5	
	ASV278	430	Snodgrassella alvi wkB298	MEIK01000026	95.6	
	ASV307	432	Snodgrassella alvi wkB2 T	CP007446	93.1	
	ASV310	431	Snodgrassella alvi wkB298	MEIK01000026	93.7	
Notes:

1) The ASVs showing <98.7% homology against the top-hit taxon are bold.

2) The superscript T means type strains of the bacteria species.

The result of the NMDS plot of β-diversity of the hindgut microbiota at the ASVs level based on Bray-Curtis dissimilarity is shown in Fig. 2. Pairwise comparisons among the three sampling periods revealed a significant difference in hindgut microbiota composition only between the BO and OW groups (F = 3.037, R2 = 0.121, p = 0.029; Table S7).

Figure 2 Nonmetric multidimensional scaling (NMDS) ordination plots of hindgut microbiota of the Japanese honey bees (Apis cerana japonica) at three sampling periods.

The plot was generated with the Bray–Curtis dissimilarity index based on the ASVs obtained from each sample. BO, before overwintering; OW, during overwintering; and AO, after overwintering.

Comparison of the core genera among the three periods

The GLMM analysis revealed that the OW group had a significant positive effect on the read counts of Bifidobacterium, Bombilactobacillus, and Lactobacillus (coefficients: 0.977, 1.036, and 0.320; 95% CI [0.237–1.716], [0.138–1.933], and [0.131–0.509]; p = 0.009, 0.024, and 0.001, respectively; Table S8).

Data availability

The raw amplicon sequence datasets generated in this study are available in the DDBJ Sequence Read Archive (accession numbers: DRR685263–DRR685298 for DRA Run and PRJDB20791 for BioProject). All scripts and datasets are deposited to figshare under DOI: 10.6084/m9.figshare.29396408.

Discussion

This study revealed that the hindgut microbiota of Apis cerana japonica was dominated by four phyla: Actinobacteriota, Bacteroidota, Firmicutes, and Proteobacteria, and six core bacterial genera: Apibacter, Bifidobacterium, Bombilactobacillus, Gilliamella, Lactobacillus, and Snodgrassella. This finding is consistent with previous studies on the gut microbiota of honey bees (Kwong et al., 2017; Dong et al., 2020). In contrast, compared with the core gut microbiota of A. mellifera, Apibacter is more abundant in Asian honey bee species, such as A. cerana, A. dorsata, and A. andreniformis (Kwong & Moran, 2016; Kwong et al., 2017; Duong et al., 2020; Ellegaard et al., 2020; Khan et al., 2023). The hindgut microbiota of Apis cerana japonica showed a similar trend at the genus level, suggesting that host phylogeny influenced microbial community structure. However, the identification of ASVs with similarities lower than the threshold for distinguishing closely related species suggests the presence of many potentially novel bacterial species, despite their genus-level similarity. Further studies involving bacterial isolation, biochemical characterization, and genome analysis are warranted to elucidate the taxonomy and function of these candidate novel bacteria.

The hindgut microbiota composition of A. c. japonica in the OW group differed significantly from that of the BO group. Notably, the mean relative abundance of Bifidobacterium, Bombilactobacillus, and Lactobacillus in OW group was higher than that in BO group. These three core bacterial genera are known to produce SCFAs from pollen-derived polysaccharides and nectar-derived glucose (Zheng et al., 2017, 2019). Among the SCFAs derived from honey bee gut bacteria, butyrate is absorbed into the hemolymph via the ileum or rectum and is therefore considered an important energy source for thermogenesis to maintain hive temperature during the overwintering period (Den Besten et al., 2013; Zheng et al., 2017). Moreover, genera Bifidobacterium and Lactobacillus contribute substantially to infection control and immune regulation in honey bees through mechanisms such as antimicrobial activity against pathogens (Wu et al., 2013) and upregulating antimicrobial peptide expression (Daisley et al., 2020). Therefore, the genus-level increase in Bifidobacterium, Bombilactobacillus, and Lactobacillus in the OW group may play a beneficial role in the overwintering of honey bees in terms of thermogenesis and immune activation. This study has limitations, as it did not experimentally evaluate the SCFAs-producing capacities or the immunomodulatory effects of Bifidobacterium, Bombilactobacillus, and Lactobacillus. Further studies quantifying SCFA levels in the gut and analyzing the expression of immune-related genes during OW are necessary to clarify the functional roles of these gut bacteria in successful overwintering.

The observed compositional changes in the hindgut microbiota of Apis cerana japonica in the overwintering period are intriguing. A possible contributing factor is the difference in pollen and nectar sources consumed by honey bees before and during overwintering period. Honey bees forage across a wide temperature range (10–40 °C) (Abou-Shaara et al., 2017), but during the overwintering period, when temperatures fall below 10 °C, they rarely leave the hive to forage (Joshi & Joshi, 2010). Consequently, honey bees are more likely to consume stored pollen and honey during overwintering period. Furthermore, the consumption of aged or stored pollen and honey can influence gut microbiota composition through changes in physiological parameters (Maes et al., 2016). In our study, although daily maximum temperatures exceeded 10 °C on all sampling days before and after overwintering period, only one-third of the days during the overwintering period reached this threshold (Japan Meteorological Agency, 2025). Another factor that may influence hindgut microbiota is variations in hive temperature. Typically, the hive temperature is maintained at 33–35.5 °C (Abou-Shaara et al., 2017). In A. c. japonica, the average winter hive temperature is 30.7 °C, while the average temperature before and after winter is 34.3 °C (Akimoto, 2000). This temperature fluctuation may affect bacterial growth rates, thereby altering microbiota composition (Ludvigsen et al., 2015; Kešnerová et al., 2017).

It is important to note that slight differences in microbiota composition were observed among colonies. Differences in gut microbiota between colonies have been reported to be influenced by the diet collected from habitat-specific floral sources (Castelli et al., 2022; Ricigliano, Williams & Oliver, 2022) and host genotype (Bridson et al., 2022). Our findings highlight the necessity of considering variation among colonies when evaluating seasonal changes in the hindgut microbiota of honey bees.

Conclusions

This study on the hindgut microbiota of A. c. japonica revealed the influence of phylogeny on microbiota composition, the presence of potentially novel species, and distinctive compositional changes during the overwintering period. The biochemical properties of the genera that increased during overwintering period (i.e., genera Bifidobacterium, Bombilactobacillus, and Lactobacillus) suggest that these changes supply energy for thermogenesis and activate the host immune system. Further surveys in other regions with different dietary environments and studies focusing on elucidating the functional roles of hindgut microbiota during overwintering and their symbiotic relationship with host health are warranted.

Supplemental Information

Supplemental Information 1 Supplementary figure and tables.

Supplemental Information 2 Raw forward sequence data of X12-3.

Supplemental Information 3 Raw forward sequence data of X3-1.

Supplemental Information 4 Raw forward sequence data of X12-1.

Supplemental Information 5 Raw reverse sequence data of X12-2.

Supplemental Information 6 Raw forward sequence data of X10-1.

Supplemental Information 7 Raw forward sequence data of X3-3.

Supplemental Information 8 Raw reverse sequence data of X3-1.

Supplemental Information 9 Raw reverse sequence data of X10-1.

Supplemental Information 10 Raw reverse sequence data of X12-1.

Supplemental Information 11 Raw forward sequence data of X3-2.

Supplemental Information 12 Raw forward sequence data of X10-3.

Supplemental Information 13 Raw reverse sequence data of X10-3.

Supplemental Information 14 Raw reverse sequence data of X10-2.

Supplemental Information 15 Raw reverse sequence data of X3-2.

Supplemental Information 16 Raw reverse sequence data of X3-3.

Supplemental Information 17 Raw forward sequence data of X12-2.

Supplemental Information 18 Raw forward sequence data of X10-2.

Supplemental Information 19 Raw reverse sequence data of X12-3.

Supplemental Information 20 Raw reverse sequence data of T10-2.

Supplemental Information 21 Raw reverse sequence data of T3-2.

Supplemental Information 22 Raw reverse sequence data of T12-1.

Supplemental Information 23 Raw reverse sequence data of T3-1.

Supplemental Information 24 Raw reverse sequence data of T3-3.

Supplemental Information 25 Raw reverse sequence data of T10-1.

Supplemental Information 26 Raw forward sequence data of T10-1.

Supplemental Information 27 Raw forward sequence data of T10-2.

Supplemental Information 28 Raw forward sequence data of T12-1.

Supplemental Information 29 Raw forward sequence data of T12-2.

Supplemental Information 30 Raw forward sequence data of T3-2.

Supplemental Information 31 Raw forward sequence data of T10-3.

Supplemental Information 32 Raw reverse sequence data of T12-2.

Supplemental Information 33 Raw reverse sequence data of T12-3.

Supplemental Information 34 Raw reverse sequence data of T10-3.

Supplemental Information 35 Raw forward sequence data of T12-3.

Supplemental Information 36 Raw forward sequence data of T3-1.

Supplemental Information 37 Raw forward sequence data of T3-3.

Supplemental Information 38 Raw forward sequence data of I3-1.

Supplemental Information 39 Raw reverse sequence data of I12-3.

Supplemental Information 40 Raw forward sequence data of I12-3.

Supplemental Information 41 Raw forward sequence data of I3-2.

Supplemental Information 42 Raw reverse sequence data of I12-1.

Supplemental Information 43 Raw reverse sequence data of I10-3.

Supplemental Information 44 Raw reverse sequence data of I10-1.

Supplemental Information 45 Raw forward sequence data of I3-3.

Supplemental Information 46 Raw reverse sequence data of I3-2.

Supplemental Information 47 Raw forward sequence data of I10-3.

Supplemental Information 48 Raw reverse sequence data of I3-1.

Supplemental Information 49 Raw forward sequence data of I10-1.

Supplemental Information 50 Raw reverse sequence data of I3-3.

Supplemental Information 51 Raw forward sequence data of I12-2.

Supplemental Information 52 Raw reverse sequence data of I12-2.

Supplemental Information 53 Raw forward sequence data of I10-2.

Supplemental Information 54 Raw reverse sequence data of I10-2.

Supplemental Information 55 Raw forward sequence data of I12-1.

Supplemental Information 56 Raw forward sequence data of H10-1.

Supplemental Information 57 Raw forward sequence data of H10-2.

Supplemental Information 58 Raw forward sequence data of H12-2.

Supplemental Information 59 Raw reverse sequence data of H3-2.

Supplemental Information 60 Raw reverse sequence data of H12-1.

Supplemental Information 61 Raw reverse sequence data of H12-3.

Supplemental Information 62 Raw reverse sequence data of H10-2.

Supplemental Information 63 Raw forward sequence data of H3-3.

Supplemental Information 64 Raw forward sequence data of H10-3.

Supplemental Information 65 Raw forward sequence data of H3-1.

Supplemental Information 66 Raw forward sequence data of H12-1.

Supplemental Information 67 Raw reverse sequence data of H10-1.

Supplemental Information 68 Raw reverse sequence data of H10-3.

Supplemental Information 69 Raw reverse sequence data of H3-3.

Supplemental Information 70 Raw reverse sequence data of H12-2.

Supplemental Information 71 Raw forward sequence data of H3-2.

Supplemental Information 72 Raw forward sequence data of H12-3.

Supplemental Information 73 Raw reverse sequence data of H3-1.

Additional Information and Declarations

Competing Interests

The authors declare that they have no competing interests.

Author Contributions

Akihiko Suzuki conceived and designed the experiments, performed the experiments, analyzed the data, prepared figures and/or tables, authored or reviewed drafts of the article, acquired funding, and approved the final draft.

Shumpei Hisamoto analyzed the data, authored or reviewed drafts of the article, and approved the final draft.

Yoshiko Sakamoto conceived and designed the experiments, authored or reviewed drafts of the article, supervised the project, and approved the final draft.

DNA Deposition

The following information was supplied regarding the deposition of DNA sequences:

The raw amplicon sequence datasets generated in this study are available in the DDBJ Sequence Read Archive (accession numbers: DRR685263–DRR685298 for DRA Run and PRJDB20791 for BioProject).

Data Availability

The following information was supplied regarding data availability:

All scripts and datasets are available at figshare: Suzuki, Akihiko; Hisamoto, Shumpei; Sakamoto, Yoshiko (2025). Dynamics of the hindgut microbiota of the Japanese honey bees (Apis cerana japonica) throughout the overwintering period. figshare. Dataset. https://doi.org/10.6084/m9.figshare.29396408.v1.

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
