# Peer review of "Dynamics of the hindgut microbiota of the Japanese honey bees (Apis cerana japonica) throughout the overwintering period"

_PeerJ, doi:10.7717/peerj.20050_

## Round 0.1 · original submission · Minor Revisions

Dear Authors, could you address the concerns of the two reviewers?

·

Basic reporting

• The manuscript is written in clear and professional English. Minor edits could improve clarity, particularly in lines such as 23, 45, and 63.
• The literature review is comprehensive and current, including recent studies (2024–2025). Comparative studies on ecological/geographic distribution of Apis cerana japonica could further strengthen the context.
• Figures and tables are of high quality and appropriately labeled. Figures 1 and 2 are especially effective.
• Raw data have been shared via DRA and FigShare, with sufficient metadata provided.

Experimental design

• The research question is clearly defined and relevant, focusing on microbiota changes in overwintering Apis cerana japonica.
• Sample collection from four colonies across three time points (BO, OW, AO) ensures systematic representation.
• High-throughput sequencing using MiSeq and analysis via DADA2, SILVA DB, and GLMM are robust and properly described.
• The study design is methodologically sound and ethically appropriate.

Validity of the findings

• The dataset is strong in sample size (n=360) and depth (coverage >99%).
• Statistical analyses (NMDS, PERMANOVA, GLMM) are well-executed and support conclusions.
• Functional implications of bacterial shifts are plausible but speculative, as no SCFA or gene expression data are presented.
• Conclusions are generally supported by data, with proper limitations discussed.

Additional comments

Strengths:
• Novel investigation on Apis cerana japonica with ecological and phylogenetic relevance.
• Identification of potential novel species is a significant contribution.
Areas for Improvement:
1. Add functional validation or explicitly state as a limitation.
2. Discuss inter-colony variability more thoroughly.
3. Deepen discussion on ecological factors (e.g., pollen/honey quality) with caution regarding speculative claims.

·

Basic reporting

Some scientific and linguistic corrections have been made, and the file is attached.

Experimental design

No comment

Validity of the findings

No comment

Additional comments

Corrections are attached.

---

## Round 0.2 · accepted · Accept

The authors have addressed all of the reviewers' comments